# Identification of a Novel Antibacterial Function of Mammalian Calreticulin

**DOI:** 10.3390/biom15070966

**Published:** 2025-07-04

**Authors:** Yichao Ma, Jiachen Liu, Xinming Qin, Xiaojing Cui, Qian Yang

**Affiliations:** MOE Joint International Research Laboratory of Animal Health and Food Safety, College of Veterinary Medicine, Nanjing Agricultural University, Weigang 1, Nanjing 210095, China; 2020207028@stu.njau.edu.cn (Y.M.);

**Keywords:** calreticulin, *Pichia pastoris*, microbial-binding, bacterial agglutination, antibacterial immunity

## Abstract

Calreticulin is a highly conserved and multifunctional molecular chaperone ubiquitously expressed in humans and animals. Beyond its well-established roles in calcium homeostasis, protein folding, and immune regulation, recent studies in aquatic species have suggested a previously unrecognized antimicrobial function of calreticulin. These findings raise the question of whether calreticulin also exerts antibacterial activity in terrestrial mammals, which has not been systematically investigated to date. To address this knowledge gap, we successfully constructed and expressed recombinant goat calreticulin using the *Pichia pastoris* expression system, yielding a protein of over 99% purity that predominantly exists in dimeric form. Functional assays demonstrated that both recombinant goat and human calreticulin exhibited preliminary inhibitory activity against *Escherichia coli*, *Salmonella typhimurium*, and *Pasteurella multocida*. Calreticulin was capable of binding to these three bacterial species as well as bacterial lipopolysaccharides (LPS). Notably, in the presence of Ca^2+^, calreticulin induced bacterial aggregation, indicating a potential mechanism for limiting bacterial dissemination and proliferation. Given the high anatomical, genetic, and physiological similarity between goats and humans—particularly in respiratory tract structure and mucosal immune function—neonatal goats were selected as a relevant model for evaluating the in vivo antimicrobial efficacy of calreticulin. Accordingly, we established an intranasal infection model using *Pasteurella multocida* to assess the protective role of calreticulin against respiratory bacterial challenge. Following infection, calreticulin expression was markedly upregulated in the nasal mucosa, trachea, and lung tissues. Moreover, intranasal administration of exogenous calreticulin significantly alleviated infection-induced pathological injury to the respiratory system and effectively decreased bacterial loads in infected tissues. Collectively, this study systematically elucidates the antimicrobial activity of calreticulin in a mammalian model and highlights its potential as a natural immune effector, providing novel insights for the development of host-targeted antimicrobial strategies.

## 1. Introduction

Calreticulin is a highly conserved and multifunctional calcium-binding molecular chaperone with an approximate molecular weight of 46 kDa. Calreticulin is structurally composed of three primary domains: a globular N-terminal domain with lectin-like properties, a proline-rich P-domain, and an acidic C-terminal domain responsible for Ca^2+^ binding. Calreticulin is predominantly localized in the endoplasmic reticulum (ER), but it can also be expressed in the cytoplasm and on the cell membrane [1,2,3]. More than 50% of intraluminal Ca^2+^ storage in the ER is associated with calreticulin. In addition to its high-affinity calcium-binding sites that enable rapid calcium binding and releasing, calreticulin interacts with calcium channels and pump-associated proteins to regulate intracellular calcium signaling pathways [4,5]. Calreticulin also functions in cooperation with various ER-resident chaperones, such as protein disulfide isomerase (PDI), to facilitate proper protein folding and assembly, thereby preventing misfolding and aggregation [6,7]. Beyond its roles in the ER, calreticulin is involved in non-ER compartment biological processes, including apoptosis [8], cell adhesion [9], antigen presentation [10], and antitumor immunity [11]. Notably, recent studies on aquatic organisms have revealed a novel antimicrobial role of calreticulin. Calreticulin homologs from *Takifugu obscurus* [12], *Paralichthys olivaceus* [13], *Amphioxus* [14], *Patinopecten yessoensis* [15], *Chlamys farreri* [16], and *Eriocheir sinensis* [17] have been shown to exert direct antibacterial and agglutination-promoting effects by binding to pathogen-associated molecular patterns (PAMPs) on bacterial surfaces, such as LPS and peptidoglycan (PGN), thereby participating in the host’s antibacterial immune response. However, whether calreticulin in mammals possesses similar antimicrobial functions remains largely unexplored. This knowledge gap is particularly critical in the context of the global crisis of antibiotic resistance, where conventional antimicrobial therapies are increasingly compromised and represent a major public health challenge. Consequently, there is an urgent need to develop novel anti-infective strategies that are safe, effective, and less prone to resistance [18]. In this regard, therapeutic approaches based on host-derived proteins have emerged as a promising direction for future antimicrobial interventions. Compared with traditional antibiotics, such endogenous protein molecules offer distinct advantages, including excellent biocompatibility, high specificity, low risk of resistance development, and the ability to modulate host immune responses. These properties provide a novel avenue for achieving precise infection control [19,20]. Therefore, in-depth investigation into the antimicrobial potential of mammalian calreticulin not only expands the functional landscape of this multifunctional protein but also provides a theoretical foundation for the development of innovative protein-based antimicrobial therapeutics.

Meanwhile, respiratory tract infections caused by pathogenic microorganisms are among the most common and severe types of infections affecting both humans and animals. These infections are typically characterized by rapid transmission and high virulence, often leading to severe pneumonia, airway obstruction, and secondary infections. The resulting clinical complications not only endanger patient survival but also impose a substantial healthcare burden and economic loss [21,22,23]. It is worth pointing out that goats, as ruminant animals, exhibit a high degree of similarity to humans in terms of respiratory tract anatomy, genetic background, and physiological characteristics. For instance, the distribution of ciliated epithelium in the nasal mucosa, the secretory functions of goblet cells and submucosal glands, and the composition of immune cells in the lamina propria are all highly comparable to those in humans, making goats an ideal animal model for studying respiratory immune defense mechanisms [24,25,26,27]. *Pasteurella multocida*, a Gram-negative coccobacillus, is widespread among domestic and wild animals and is a common etiological agent of respiratory infections. In ruminants, it is particularly associated with hemorrhagic pneumonia, pulmonary hemorrhage, and septicemia, resulting in considerable economic losses [28]. Of greater concern is the zoonotic potential of this pathogen, which can cause infections in humans and may even lead to systemic septicemia in immunocompromised individuals, potentially causing life-threatening systemic infections [29]. Therefore, investigating whether calreticulin plays a role in the clearance of *Pasteurella multocida* infection is not only of practical significance for the control of animal diseases but also provides theoretical support for the treatment of human respiratory infections.

In this study, we successfully expressed and purified recombinant *Capra hircus* (goat) calreticulin using the *Pichia pastoris* expression system. The purified calreticulin exhibited potent antibacterial activity against *Escherichia coli*, *Salmonella typhimurium*, and *Pasteurella multocida*. Notably, calreticulin was found to bind directly to these bacterial pathogens as well as to the pathogen-associated molecular pattern LPS, and to induce calcium-dependent bacterial agglutination. Furthermore, intranasal administration of recombinant calreticulin to 30-day-old lambs significantly alleviated *Pasteurella multocida*-induced pathological lesions and markedly reduced bacterial burdens in infected respiratory tissues. To our knowledge, this is the first study to systematically characterize the antibacterial function of mammalian calreticulin, particularly its critical role in respiratory tract defense. These findings provide both a theoretical framework and experimental evidence for advancing our understanding of calreticulin-mediated immune regulation and for developing novel calreticulin-based anti-infective strategies.

## 2. Materials and Methods

### 2.1. Bacterial Strains, Antibodies, and Reagents

Bacterial Strains: *Pasteurella multocida* (serotype D) was cultured in Tryptic Soy Broth (TSB) supplemented with 5% fetal bovine serum (FBS) and 0.5% nicotinamide adenine dinucleotide (NAD); *Streptococcus* (ZY05719) was grown in Todd–Hewitt Broth (THB) containing 5% FBS; *Escherichia coli* (O157:H7), *Salmonella typhimurium* (ATCC 14028), and *Staphylococcus aureus* (ATCC 25923) were cultured in Luria–Bertani (LB) broth.

Antibodies: GAPDH antibody (Proteintech, Wuhan, China, 60004-1-Ig, Western blot, 1:1000); calreticulin antibody (Proteintech, 10292-1-AP, Western blot, 1:1000; IHC, 1:200); His-tag (Proteintech, 66005-1-Ig, Western blot, 1:10,000).

Reagents: SABC-POD (rabbit IgG) kit (BOSTER, Wuhan, China, SA1020); carboxyfluorescein diacetate succinimidyl ester (CFSE) (Thermo Fisher Scientific, Waltham, MA, USA, 65-0850-84); recombinant human calreticulin (Sino Biological, Beijing, China, P27797).

### 2.2. Animals

The 30-day-old goats used in this study were obtained from Nanjing Zhushun Biotechnology Co., Ltd. (Nanjing, China). The animal protocol was approved by the Animal Ethics Committee of Nanjing Agriculture University.

### 2.3. Preparation of Recombinant Calreticulin

The coding sequence of the calreticulin gene was initially cloned into the expression vector pPIC9K based on homologous recombination technology using the Vazyme ClonExpress II One Step Cloning Kit (Vazyme, Nanjing, China, C112). The recombinant plasmid PpIC9K-calreticulin was then electroporated into *Pichia pastoris* GS115 competent cells. Positive transformants were selected on minimal dextrose (MD) agar plates lacking histidine (His^−^). Individual colonies were inoculated into test tubes containing buffered glycerol complex medium (BMGY) and cultured at 30 °C with shaking at 220 rpm. Then, when the culture reached an OD_600_ of 3.0, the cells were moved to buffered methanol complex medium (BMMY) to induce recombinant protein expression, with methanol added to a final concentration of 0.5% (*v*/*v*) every 24 h to maintain induction. On the third day of induction, 1 mL of culture was harvested and centrifuged at 12,000 rpm for 10 min at 4 °C. The supernatant was collected, mixed with 5× SDS loading buffer, boiled for 15 min, and analyzed by SDS-PAGE and Western blotting to verify the expression of recombinant calreticulin. Following successful expression verification, recombinant calreticulin was purified from the culture supernatant using His-tag affinity column (HisSep Ni-NTA 6FF, 5 mL) chromatography. The eluted protein was then desalted using a HiTrap™ Desalting column (5 mL) and buffer-exchanged into phosphate-buffered saline (PBS, 0.01 M, pH 7.2) for subsequent applications.

### 2.4. Antibacterial Activity Assay of Calreticulin

*Pasteurella multocida*, *Escherichia coli*, *Salmonella typhimurium*, *Streptococcus*, and *Staphylococcus aureus* were cultured to logarithmic phase and diluted in Tris-buffered saline (TBS) to a final concentration of 1 × 10^5^ colony-forming units (CFU) per 12.5 μL. The bacterial suspensions were then incubated with 50 μL of recombinant calreticulin at a final concentration of 100 μg/mL, prepared in Tris-buffered saline (TBS) with or without 10 mM Ca^2+^. The concentration of recombinant calreticulin (100 μg/mL) and calcium (10 mM) was selected based on previous studies that demonstrated in vitro biological activity of calreticulin at comparable levels in aquatic species [12,13]. The mixtures were incubated at 37 °C for 3 h. Bovine serum albumin (BSA, 100 μg/mL) served as the negative control. After incubation, either the entire reaction mixture or appropriate serial dilutions were plated onto selective solid agar plates appropriate for each bacterial strain and incubated at 37 °C for 16–18 h to allow for colony formation. The number of CFUs was subsequently counted to evaluate the antibacterial efficacy of calreticulin.

### 2.5. Determination of Bacterial Growth Curves

*Escherichia coli*, *Salmonella typhimurium*, and *Pasteurella multocida* were cultured to logarithmic phase, then diluted with fresh medium to a final concentration of 1 × 10^6^ CFU/mL. Recombinant calreticulin was administered to the treatment group at a final concentration of 100 μg/mL. Control groups received supplementation with either bovine serum albumin (BSA, 100 μg/mL) or an equivalent volume of buffer, both containing 10 mM Ca^2+^. All samples were incubated at 37 °C with shaking at 180 rpm. At 0, 2, 4, 6, 8, 10, and 12 h; aliquots were taken from each group; and bacterial growth was assessed by measuring the OD_600_. Growth curves were constructed by plotting OD_600_ values against time (h) to assess the bacteriostatic effect of calreticulin.

### 2.6. Calreticulin–Bacteria-Binding Assays

The binding ability of calreticulin to bacteria or LPS (extracted from *Escherichia coli* O55:B5; Sigma-Aldrich; St. Louis, MO, USA; Catalog No. L6529) was determined by enzyme-linked immunosorbent assay (ELISA) as reported previously [30,31,32].

ELISA assay: *Pasteurella multocida*, *Escherichia coli*, and *Salmonella typhimurium* were diluted in coating buffer to a final concentration of 1 × 10^7^ CFU/mL, and 100 μL per well was added to 96-well ELISA plates, followed by incubation at 4 °C overnight for immobilization. The plates were then blocked with 5% BSA at room temperature for 1.5 h. After blocking, the following reagents were added to designated wells: Ca^2+^, His-tag peptide, His-tag + Ca^2+^, calreticulin, and calreticulin + Ca^2+^. Plates were incubated at 37 °C for 2 h, followed by three washes with TBST (Tris-buffered saline with 0.05% Tween-20). HRP-conjugated anti-His monoclonal antibody was then added and incubated for 1 h at room temperature. After washing again with TBST, TMB substrate was introduced and incubated for 15 min. The reaction was terminated by the addition of 2 mol/L H_2_SO_4_, and absorbance was measured at 450 nm using a microplate reader.

For the LPS binding assay, LPS was coated on the ELISA plates at 40 μg/mL, and the remaining steps were performed as described above for the bacterial binding ELISA.

Western blot assay: Logarithmic-phase *Escherichia coli*, *Salmonella typhimurium*, and *Pasteurella multocida* were washed with sterile PBS, and resuspended to a final concentration of 2 × 10^7^ CFU/mL. Bacterial suspensions (500 μL) were incubated with an equal volume of purified calreticulin (100 μg/mL) at room temperature for 1 h on a rotator. Bacteria incubated with 100 μg/mL His-tag peptide served as negative controls. After incubation, bacteria were washed three times with PBS, and surface-bound proteins were eluted using 5% SDS. The presence of calreticulin bound to bacteria was analyzed by Western blot.

### 2.7. Western Blot Analysis

Protein samples were mixed with 5× SDS loading buffer and denatured by boiling in a water bath for 15 min. Subsequently, proteins were separated by SDS-PAGE and transferred onto methanol-activated PVDF membranes using a Bio-Rad transblot apparatus. The membranes containing target proteins were blocked with 5% non-fat milk in TBST at room temperature for 2 h. After blocking, the membranes were incubated overnight at 4 °C with primary calreticulin antibody and GAPDH antibody. The membranes were then washed five times with TBST (5 min per wash). HRP-conjugated goat anti-rabbit secondary antibody was applied and incubated at room temperature for 1.5 h. After additional washes with TBST, the protein bands were visualized using enhanced chemiluminescence (ECL) substrate and imaged with an automated chemiluminescence imaging system. Densitometric analysis was performed using ImageJ software (version 1.8.0, Bethesda, MD, USA).

### 2.8. Bacterial Agglutination Assay

Logarithmic-phase cultures of *Escherichia coli*, *Salmonella typhimurium*, and *Pasteurella multocida* were collected through centrifugation at 5000 rpm for 10 min at 4 °C. The bacterial pellets were washed three times with sterile PBS and subsequently labeled with CFSE. To ensure appropriate controls and evaluate the specific effects of calreticulin and calcium, five experimental groups were included: (1) Ca^2+^ only group, to assess the effect of calcium ions alone; (2) His-tag group, to control for potential effects of the recombinant tag; (3) His-tag + Ca^2+^ group, to exclude non-specific effects of the tag in the presence of calcium; (4) calreticulin group, to examine the activity of calreticulin without calcium; and (5) calreticulin + Ca^2+^ group, to evaluate the agglutination-inducing activity of calreticulin under calcium-supplemented conditions. CFSE-labeled bacteria were incubated with calreticulin (100 μg/mL) in the presence or absence of 10 mM Ca^2+^ at 37 °C for 2 h. Following incubation, bacterial agglutination was assessed using a fluorescence microscope.

### 2.9. Histopathological Detection

Paraffin-embedded tissue sections were sequentially deparaffinized in xylene and rehydrated with gradient ethanol (100%, 95%, 85%, and 70%), followed by tap water rinsing. Finally, after dehydration with gradient ethanol and permeabilization with xylene, the slides were sealed with neutral gum. After complete air-drying, the slides were observed and photographed under a light microscope.

### 2.10. Immunohistochemical Staining

Paraffin-embedded tissue sections were deparaffinized in xylene and rehydrated through a graded ethanol series (100%, 95%, 85%, and 75%). Antigen repair was performed by incubating the sections in citrate-citrate sodium buffer (pH = 6.0) for 30 min. Endogenous peroxidase activity was eliminated by incubation with 3% hydrogen peroxide at 37 °C for 1 h. After three washes with PBS, sections were blocked with 5% BSA for 1.5 h to prevent nonspecific antibody binding, followed by incubation with primary calreticulin antibody at 4 °C overnight. After washing again three times with PBS, biotinylated secondary antibody was added for 1.5 h at room temperature. Following the washing step, streptavidin–biotin complex (SABC) reagent was introduced and incubated at 37 °C for 1 h. Color development was performed utilizing 3,3′-diaminobenzidine (DAB) substrate, with the dyeing time monitored by microscopic visualization. Finally, the nuclei were counterstained with hematoxylin, and the sections were dehydrated, cleared, and mounted for microscopic observation.

### 2.11. In Vivo Evaluation of Calreticulin’s Protective Effect Against Pasteurella multocida

Nine healthy goats with similar body weight were randomly divided into three groups (*n* = 3 per group): control group, *Pasteurella multocida* infection group, and calreticulin treatment group. Goats in the infection and calreticulin groups were intranasally challenged with *Pasteurella multocida* at a concentration of 1 × 10^8^ CFU/mL, while the control group received an equivalent volume of PBS buffer via the same route. At 6 h post-infection, the calreticulin treatment group was administered 1 mL of calreticulin solution (2.5 mg/mL) intranasally; the control and infection groups received the same volume of calreticulin vehicle buffer (TBS buffer containing 10 mM Ca^2+^). At 24 h post-infection, all goats were euthanized humanely in accordance with institutional animal care guidelines. The euthanasia time point was predetermined based on the onset of consistent and marked clinical symptoms observed in infected animals, including hyperthermia (>40.5 °C), labored breathing, nasal discharge, coughing, lethargy, and decreased appetite, which were observed within 12–18 h post-infection and progressed consistently across individuals. These signs indicated systemic infection and the onset of animal distress. Therefore, to avoid prolonged suffering and ensure reliable assessment of high-quality respiratory tissues for subsequent microbiological and histological evaluations, tissue collection was uniformly scheduled at 24 h post-infection.

### 2.12. RT-qPCR

Total RNA was extracted from goat tissues using the Trizol method. RNA concentration and purity were measured with an ultraviolet spectrophotometer, and RNA samples were adjusted to 500 ng/μL. cDNA was reverse-transcribed using the HiScript II Reverse Transcriptase kit (Vazyme, Nanjing, China, R201). Quantitative real-time PCR (qPCR) was then conducted according to the manufacturer’s instructions for the ChamQ SYBR qPCR Master Mix (Vazyme, Nanjing, China, Q311). Primer sequences are listed in Appendix A. Analyses of relative gene expression were determined using the 2^−ΔΔCt^ method with GAPDH as the internal reference gene.

### 2.13. Statistical Analysis

Data were analyzed using GraphPad Prism version 9.4.1, and results are presented as means ± standard deviation (SD). One-way analysis of variance (ANOVA) was used to assess significant differences among multiple groups, while *t*-tests were applied to evaluate differences between two groups. Significance thresholds were set as follows: NS—not significant; * *p* < 0.05; ** *p* < 0.01; *** *p* < 0.001.

## 3. Results

### 3.1. Evolutionary Conservation and Sequence Homology Analysis of Calreticulin Across Different Species

To elucidate the evolutionary conservation and potential functional diversification of calreticulin across different taxonomic groups, we retrieved and aligned the full-length amino acid sequences of calreticulin from 40 representative species, including mammals, birds, reptiles, amphibians, and fish. A maximum likelihood phylogenetic tree was constructed based on these sequences and is presented in Figure 1A. The phylogenetic analysis revealed that calreticulin is highly conserved among mammals. Species such as *Homo sapiens*, *Pan troglodytes*, *Mus musculus*, *Capra hircus*, *Canis lupus familiaris*, *Panthera tigris*, and *Hippopotamus amphibius* clustered closely within a major clade at the proximal end of the tree. Most nodes within this clade displayed high bootstrap values (>80), indicating strong sequence homology and evolutionary stability. In contrast, calreticulin sequences from non-mammalian vertebrates—including birds (e.g., *Gallus gallus*, *Coturnix japonica*, *Taeniopygia guttata*), reptiles (e.g., *Pogona vitticeps*, *Eublepharis macularius*), and teleost fish (e.g., *Salmo salar*, *Cyprinus carpio*, *Betta splendens*)—were grouped into distinct and relatively divergent clades. Furthermore, sequence alignment analysis of calreticulin from multiple species revealed a high degree of sequence conservation (Figure 1B). For instance, the amino acid sequence homology of goat calreticulin compared to human, mouse, pig, and primates (*Mandrillus leucophaeus*) was 92.90%, 92.40%, 96.20%, and 93.10%, respectively. Taken together, this strong sequence conservation and evolutionary stability support the notion that calreticulin may carry evolutionarily conserved physiological functions that have been stably maintained throughout evolution. However, its potential role in antibacterial defense requires further experimental validation.

### 3.2. Expression Analysis of Calreticulin in Goat Tissues

Immunohistochemical analysis demonstrated that, under physiological conditions, calreticulin is ubiquitously expressed in a range of goat tissues, including the nasal cavity, pharynx, trachea, lung, heart, liver, spleen, and intestinal tissues (Figure 2A). In the respiratory mucosa, calreticulin was abundantly expressed in the submucosal glands of the nasal cavity, glands of the pharynx, submucosal glands of the trachea, as well as in the epithelial cells of the bronchioles and alveoli in lung tissue. In cardiac tissue, calreticulin was broadly distributed in both cardiomyocytes and vascular endothelial cells. It was also widely expressed in hepatocytes within the liver parenchyma. Within the spleen, calreticulin expression was detected in both the white pulp and red pulp. Additionally, calreticulin was abundantly expressed in the intestinal epithelial tissue. Western blot analysis (Figure 2B,C) further demonstrated that calreticulin expression was highest in respiratory mucosa (nasal cavity, pharynx, trachea, and lung), followed by the intestine and spleen, with lower expression levels observed in the heart and liver. Taken together, calreticulin is broadly expressed across multiple goat tissues, with the highest abundance in the respiratory mucosa, suggesting a potential critical role in the mucosal immune defense of the respiratory tract.

### 3.3. Expression and Purification of Goat Calreticulin

In this study, the full-length calreticulin gene sequence was cloned into the PpIC9K expression vector (Figure 3A,B). The recombinant PpIC9K-calreticulin plasmid was confirmed by single and double enzyme digestion. Positive transformants were induced for expression, and the yeast culture supernatants were collected for SDS-PAGE and Western blot analysis. The supernatants from expression-positive clones were subjected to Ni-NTA affinity chromatography purification. As shown in Figure 3C,D, compared to the uninduced culture supernatant, the methanol-induced supernatant exhibited a positive calreticulin band at approximately 60 kDa, indicating successful secretory expression of the recombinant protein with high purity. Furthermore, size exclusion high-performance liquid chromatography (SEC-HPLC) was utilized to compare the yeast-secreted recombinant goat calreticulin with recombinant human calreticulin expressed in HEK293 cells. As shown in Figure 3E, both recombinant proteins exhibited single peaks at 7.950 min and 7.859 min, respectively. Compared to the retention time of the standard (Appendix A), these results indicate that both recombinant proteins predominantly exist in a dimeric form, with a purity of 99%. These data demonstrate the successful construction and high-purity expression of recombinant goat calreticulin, predominantly in a dimeric form, providing a basis for further functional studies.

### 3.4. Verification of Antibacterial Activity of Calreticulin

The bactericidal activity of recombinant goat and human calreticulin was evaluated. As shown in Figure 4A–C and Appendix A, both proteins (in the presence of 10 mM Ca^2+^) exhibited bactericidal effects against *Escherichia coli*, *Salmonella typhimurium*, and *Pasteurella multocida*. Compared to the control group, treatment with calreticulin reduced the CFUs of *Escherichia coli*, *Salmonella typhimurium*, and *Pasteurella multocida*, respectively. In contrast, calreticulin from both species showed no inhibitory effect on *Streptococcus* and *Staphylococcus aureus* (Appendix A). To further explore the concentration dependence of this activity, we tested higher doses of calreticulin. As shown in Appendix A, treatment with 1000 μg/mL of recombinant calreticulin resulted in a reduction in CFUs by approximately 1.69 ± 0.07 log for *Escherichia coli*, 2.61 ± 0.20 log for *Salmonella typhimurium*, and 2.29 ± 0.22 log for *Pasteurella multocida*, compared to the untreated control group. Next, we assessed the impact of recombinant goat calreticulin on bacterial growth curves. As illustrated in Figure 4D–F, bacterial growth of *Escherichia coli*, *Salmonella typhimurium*, and *Pasteurella multocida* was suppressed in the calreticulin-treated groups compared with the BSA-treated and blank control groups. A similar concentration-dependent inhibitory trend was observed in the growth curve analyses (Appendix A), where increased calreticulin levels were associated with delayed bacterial proliferation. These data provide preliminary evidence that calreticulin exerts a dose-dependent inhibitory effect on the growth of Gram-negative bacteria under calcium-enriched conditions. In contrast, no inhibitory activity was detected against Gram-positive bacteria.

### 3.5. Calreticulin Mediates Bacterial Binding and Aggregation

The binding interaction between calreticulin and *Escherichia coli*, *Salmonella typhimurium*, and *Pasteurella multocida* was assessed using ELISA and Western blot analysis. As shown in Figure 5A–C, calreticulin was able to bind all three bacterial species, with enhanced binding observed in the presence of 10 mM Ca^2+^. Western blot results (Figure 5D) showed no signal in the His-tag-negative control eluates, while distinct ~60 kDa bands were detected in calreticulin eluates, indicating specific and effective binding. To improve the interpretability and reproducibility of the binding data, we further performed concentration–gradient binding assays using increasing doses of recombinant calreticulin (1, 10, 100, 500, and 1000 μg/mL). As shown in Appendix A, the OD_450_ values increased proportionally with calreticulin concentration, demonstrating a clear dose-dependent binding trend. Notably, the signal tended to plateau at higher concentrations (≥100 μg/mL), suggesting potential saturation of available bacterial binding sites. To explore the molecular basis underlying calreticulin–bacteria interactions, we next assessed its binding to LPS, a major component of the outer membrane of Gram-negative bacteria. As shown in Figure 5E, ELISA revealed a concentration-dependent interaction between calreticulin and LPS, indicating that LPS may serve as one of the primary ligands mediating calreticulin recognition of bacterial surfaces. Additionally, bacterial aggregation assays using CFSE-labeled bacteria revealed that *Escherichia coli*, *Salmonella typhimurium*, and *Pasteurella multocida* were evenly dispersed in the Ca^2+^, His-tag, and His-tag + Ca^2+^ groups. In contrast, significant aggregation was observed only in the calreticulin + Ca^2+^ group, indicating that calreticulin promotes bacterial aggregation in a Ca^2+^-dependent manner (Figure 5F).

### 3.6. Upregulation of Calreticulin Expression in Respiratory Mucosa Following Pasteurella multocida Infection in Lambs

Thirty-day-old lambs were intranasally infected with *Pasteurella multocida* for 24 h, resulting in a significant upregulation of calreticulin expression levels in various regions of the respiratory tract, including the nasal cavity, trachea, and lungs. As shown in Figure 6A,B, immunohistochemical staining revealed markedly increased calreticulin expression in these respiratory regions following infection. Furthermore, RT-qPCR analysis demonstrated that, compared to the control group, calreticulin mRNA levels in the nasal cavity, trachea, and lung were increased after *Pasteurella multocida* infection (Figure 6C). Consistently, Western blot analysis confirmed a corresponding increase in calreticulin protein expression in these regions (Figure 6D,E). These findings suggest that calreticulin may serve as an important defensive or regulatory molecule in the nasal mucosa in response to *Pasteurella multocida* infection.

### 3.7. Intranasal Administration of Calreticulin Alleviates Pathological Damage and Promotes Pasteurella multocida Clearance in Lambs

Pasteurellosis, caused by *Pasteurella multocida*, is a severe infectious disease characterized primarily by pneumonia and multi-organ hemorrhage, and it is a leading cause of mortality in suckling lambs. In humans, pasteurellosis typically presents as a localized wound infection following an animal bite or scratch, which may progress to severe soft tissue infection, abscess formation, septic arthritis, or osteomyelitis. Additionally, *Pasteurella multocida* can cause meningitis, ocular infections, and respiratory tract infections, particularly in patients with underlying pulmonary disease. In this study, 30-day-old lambs were intranasally infected with *Pasteurella multocida*. At 6 h post-infection, lambs were treated with calreticulin via intranasal administration to evaluate its effect on in vivo clearance of *Pasteurella multocida*. Histopathological analysis of the nasal cavity, trachea, and lungs using hematoxylin–eosin staining. As shown in Figure 7A,B, the *Pasteurella multocida*-infected group exhibited disrupted nasal epithelial integrity accompanied by congestion or hemorrhage. No significant histopathological changes were observed in the trachea across all groups. Moreover, the lungs of infected lambs showed thickening of the alveolar walls, along with congestion and hemorrhage. However, intranasal calreticulin treatment significantly alleviated the pathological damage caused by the infection. Furthermore, the distribution of *Pasteurella multocida* in lamb tissues was detected using a *Pasteurella multocida*-specific fluorescent probe. Compared to the infected group, the calreticulin-treated group showed a significant reduction in bacterial load in the nasal cavity, trachea, and lungs (Figure 7C,D). These results further confirm that calreticulin accelerates the clearance of *Pasteurella multocida* in vivo, supporting its role as a potential protective agent in respiratory mucosal immunity.

## 4. Discussion

Host endogenous proteins possess broad potential value in clinical applications, spanning several critical areas such as the treatment of infectious diseases, immunomodulatory therapies, preventive immunization, personalized medicine, and novel drug development. These proteins not only represent essential components of the host immune defense system but also play key roles in the treatment of various diseases, thereby driving the advancement of precision medicine and individualized treatment [33,34,35]. Calreticulin is a highly conserved, multifunctional calcium-binding protein identified across a broad spectrum of species, including plants, invertebrates, and vertebrates [36,37]. In aquatic animals, calreticulin has been shown to play an important role in antibacterial immune responses. However, whether calreticulin exhibits similar antibacterial functions in mammals remains insufficiently elucidated and requires further detailed investigation. In this study, we successfully obtained secretory expression of full-length goat calreticulin using the *Pichia pastoris* expression system for the first time. Functional analysis demonstrated that recombinant calreticulin effectively inhibited the growth of *Escherichia coli*, *Salmonella typhimurium*, and *Pasteurella multocida*. Additionally, intranasal administration of calreticulin effectively reduced the bacterial load of *Pasteurella multocida* in lambs.

Calreticulin is widely expressed in both vertebrates and invertebrates, and is localized to multiple subcellular compartments [38,39]. In teleosts such as *Paralichthys olivaceus*, *Takifugu obscurus*, *Branchiostoma japonicum*, *Oncorhynchus mykiss*, and *Sebastes schlegeli*, calreticulin is broadly expressed in tissues including the spleen, muscle, head kidney, brain, heart, gills, intestine, blood, and liver, with particularly high expression levels in the liver [40]. In invertebrates, such as earthworms, calreticulin is highly expressed in various cells and tissues, including the epidermis, neurons of the ventral nerve cord, intestine, sperm, body wall muscles, and some coelomocytes [41]. This study is the first to systematically evaluate the expression profile of calreticulin across different tissues in goats. Calreticulin was broadly distributed across goat tissues, with particularly high expression in the respiratory and intestinal tracts, and comparatively lower levels in the heart and liver. This tissue-specific expression pattern suggests a potential role for calreticulin in mucosal epithelial immune defense. Further histological analysis revealed pronounced expression of calreticulin in the submucosal glands of the respiratory tract, bronchial epithelial cells, and intestinal epithelial cells, suggesting that calreticulin may be secreted into body fluids and act as a soluble molecule involved in mucosal immune regulation.

The *Pichia pastoris* expression system, as an efficient and stable eukaryotic expression platform, has been widely applied in biopharmaceuticals, biochemistry, and biotechnology. Its genome contains strong promoters and secretion signal sequences, enabling high-level secretion of heterologous proteins while maintaining expression stability in large-scale fermentation. Additionally, the secretory pathway and organelle architecture of *Pichia pastoris* resemble those of mammalian cells, facilitating proper folding, glycosylation, phosphorylation, and other post-translational modifications of target proteins, thus ensuring their bioactivity and function [42,43]. In this study, we successfully constructed a *Pichia pastoris* expression vector and obtained recombinant goat calreticulin, which primarily exists in a dimeric form. The amino acid sequence of calreticulin contains intrinsically disordered regions (IDRs) lacking stable three-dimensional structures. The IDRs provide conformational flexibility, enabling calreticulin to assume a dynamic structural ensemble. Such structural characteristics may partially drive spontaneous dimerization or the formation of higher-order oligomers [44,45]. Recombinant calreticulin from species such as *Branchiostoma japonicum*, *Patinopecten yessoensis*, and *Chlamys farreri* has been demonstrated to possess strong pathogen-binding and antibacterial capabilities, effectively binding to *Staphylococcus aureus* and *Escherichia coli* and inhibiting their growth. Similarly, recombinant calreticulin from *Eriocheir sinensis* exhibits binding affinity to multiple bacteria, including *Escherichia coli*, *Aeromonas hydrophila*, *Staphylococcus aureus*, *Micrococcus luteus*, *Bacillus subtilis*, *Bacillus megaterium*, *Vibrio anguillarum*, and *Vibrio parahaemolyticus*. In addition, calreticulins from these aquatic animals can also bind to bacterial LPS and peptidoglycan (PGN). In our study, mammalian goat calreticulin was shown to bind *Escherichia coli*, *Salmonella typhimurium*, and *Pasteurella multocida* while also inhibiting their growth. Recombinant goat calreticulin demonstrated binding affinity to LPS as well. The interaction between calreticulin and pathogens or their LPS may depend on a cooperative recognition mechanism involving multiple functional domains. The P-domain of calreticulin exhibits lectin-like activity, capable of recognizing and binding various bacterial surface-exposed glycans, particularly high-mannose-type glycans and other glycosaminoglycan-rich oligosaccharides. These glycans are commonly found in the core oligosaccharide region of LPS and on certain bacterial surface glycoproteins [46,47,48]. In addition, calreticulin may recognize pathogens through non-carbohydrate-dependent mechanisms. Positively charged regions within its molecular structure could interact electrostatically with negatively charged bacterial surface molecules, such as phosphate groups in LPS or the lipid A portion. Hydrophobic residues in calreticulin may also engage with hydrophobic components of the bacterial outer membrane, further strengthening bacterial binding [49,50,51].

Notably, the antibacterial activity of calreticulin was observed exclusively in the presence of calcium ions. This calcium-dependent effect is likely attributed to the structural and functional regulation of the protein. Calreticulin is characterized by a high-affinity calcium-binding C-terminal domain, which is enriched in acidic residues such as aspartic acid and glutamic acid. Upon binding to Ca^2+^, calreticulin undergoes marked conformational transitions, shifting from a relatively disordered or intermediate-folding state into a more ordered and functionally active tertiary structure. These structural rearrangements are essential for the proper spatial exposure and orientation of functional domains involved in bacterial surface binding. Importantly, calcium-induced conformational changes also facilitate the oligomerization of calreticulin, resulting in the formation of multivalent binding platforms that can simultaneously engage multiple repetitive motifs on bacterial surfaces [52,53]. This oligomeric state enhances its capacity to mediate bacterial agglutination—a process that not only physically restricts bacterial motility and dissemination but also contributes to antibacterial activity through several mechanisms, including electrostatic shielding, structural stabilization, and enhanced hydrophobic interactions at the protein–bacteria interface. Collectively, these effects may disrupt the homeostasis of bacterial membranes, thereby inhibiting bacterial growth. Furthermore, the larger bacterial aggregates formed by calreticulin-mediated agglutination in vivo may be more readily recognized and eliminated by professional phagocytes, such as macrophages and neutrophils. In addition, calreticulin-induced bacterial clustering can spatially confine freely diffusing pathogens, thereby preventing their systemic dissemination and tissue invasion within the host.

Building upon the calcium-dependent antibacterial mechanisms observed in vitro, we further investigated whether calreticulin exerts protective effects in vivo. Previous studies have shown that in various aquatic animals, the expression levels of calreticulin are significantly upregulated following infection by multiple pathogenic bacteria, suggesting a crucial role for calreticulin in pathogen recognition and clearance [54,55]. Consistent with these findings, our study observed a notable induction of calreticulin expression in multiple respiratory tissues and organs of lambs following intranasal challenge with *Pasteurella multocida*, indicating that calreticulin similarly contributes to the immune response against bacterial infection in mammals. More importantly, exogenous supplementation of calreticulin enhances the host’s ability to clear pathogens. In this study, intranasal administration of recombinant calreticulin effectively accelerated the clearance of *Pasteurella multocida* in lambs and significantly alleviated respiratory tissue pathology caused by the infection. These results suggest that calreticulin may serve as a key defensive or regulatory molecule in the nasal mucosa against *Pasteurella multocida* infection. However, this in vivo experiment has certain limitations. Due to the acute nature of the infection model and the rapid onset of severe clinical signs, all animals were humanely euthanized at 24 h post-infection to avoid prolonged suffering and to ensure the collection of high-quality tissue samples. As a result, longitudinal clinical observations such as changes in body temperature, weight, respiratory rate, and survival could not be continuously monitored, which restricts the depth of in vivo efficacy evaluation. Therefore, although bacterial burden and histopathological improvement provide preliminary support for the therapeutic potential of calreticulin, further studies incorporating extended observation periods, multiple infection doses, and dynamic clinical scoring systems are needed to comprehensively validate its protective efficacy and safety.

## 5. Conclusions

In summary, this study is the first to systematically demonstrate the antibacterial activity of calreticulin against *Escherichia coli*, *Salmonella typhimurium*, and *Pasteurella multocida* in a mammalian model. In vitro assays demonstrated that calreticulin can bind to these pathogens and LPS and induce bacterial agglutination in the presence of Ca^2+^, thereby limiting their dissemination and proliferation. In vivo, we further confirmed that calreticulin expression is significantly upregulated following *Pasteurella multocida* infection in goats, and that intranasal supplementation of calreticulin not only alleviates tissue damage caused by bacterial infection but also significantly accelerates pathogen clearance. These findings highlight the potential of calreticulin as a host defense factor and provide new insights for the development of antimicrobial therapeutic strategies.

## Figures and Tables

**Figure 1 biomolecules-15-00966-f001:**
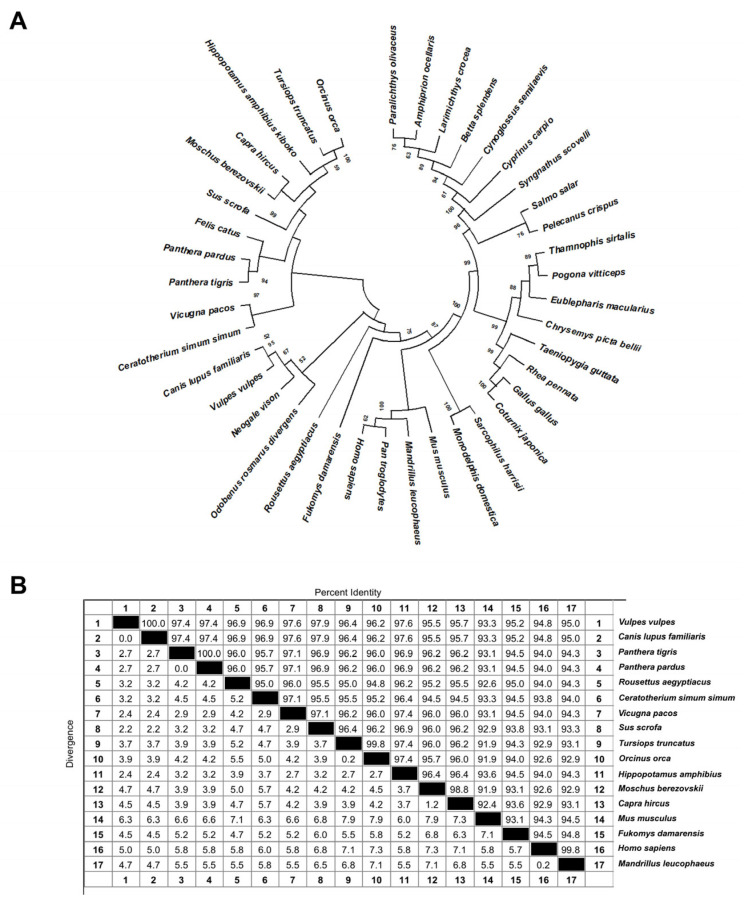
Evolutionary conservation and sequence homology analysis of calreticulin across multiple species. (**A**) Phylogenetic analysis of calreticulin amino acid sequences from different species. The neighbor-joining phylogenetic tree was constructed using the bootstrap method in MEGA version 10.2.2 with 1000 bootstrap replicates. (**B**) Sequence homology analysis of calreticulin amino acid sequences among various species. The matrix shown represents a bidirectional pairwise comparison of calreticulin sequences. The upper triangular region displays the percentage of sequence identity (% identity), while the lower triangular region shows the corresponding percentage of sequence divergence (% divergence) between each pair of species. Black squares along the diagonal indicate self-alignments, where sequence identity is 100% and divergence is 0%.

**Figure 2 biomolecules-15-00966-f002:**
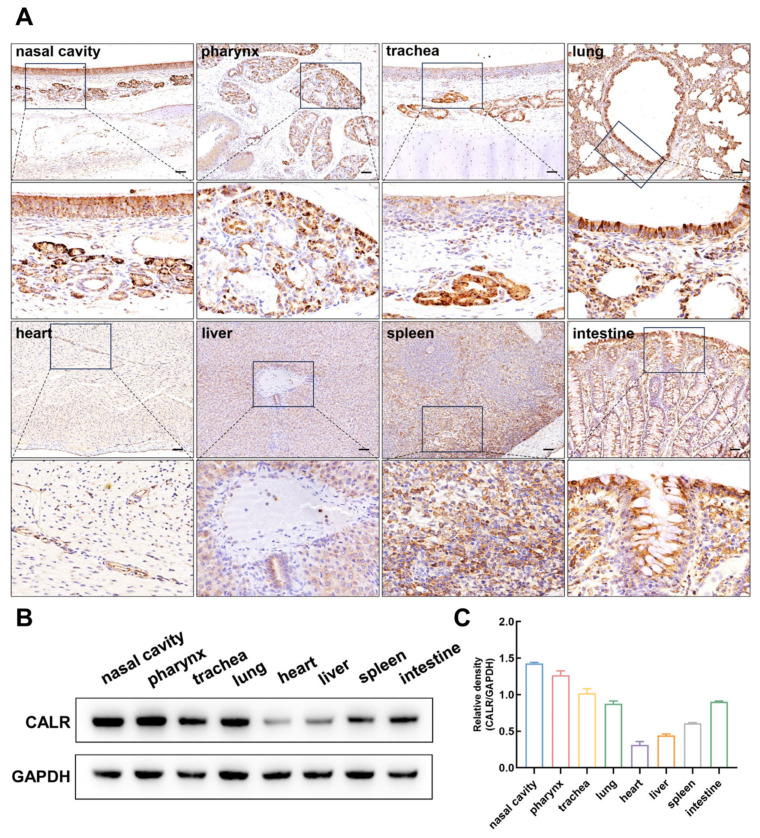
Distribution of calreticulin in different goat organs. (**A**) Immunohistochemical staining illustrating the localization of calreticulin in the nasal mucosa, pharynx, trachea, lung, heart, liver, spleen, and small intestine of goats. Bars = 50 μm. (**B**) Analysis of calreticulin protein expression levels via Western blot in the nasal mucosa, pharynx, trachea, lung, heart, liver, spleen, and small intestine of goats. (**C**) The band obtained by Western blot was analyzed by gray value, and the result was expressed as the gray value ratios of the CALR/GAPDH. All data shown are the mean ± SD from three independent experiments. Original Western blot images can be found in Appendix A.

**Figure 3 biomolecules-15-00966-f003:**
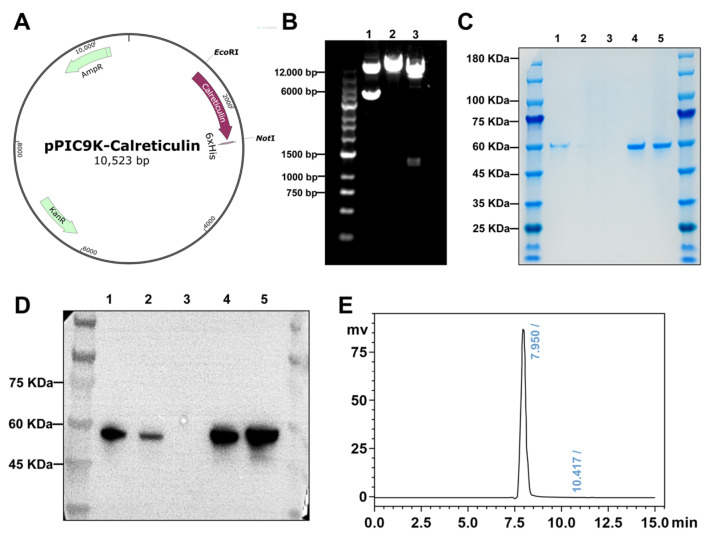
Expression and purification of recombinant goat calreticulin using the *Pichia pastoris* expression system. (**A**) Schematic representation of the recombinant plasmid PpIC9K-CALR. (**B**) Agarose gel electrophoresis was performed to analyze the original recombinant plasmid (lane 1), the plasmid digested with EcoRI alone (lane 2), and the plasmid digested with both EcoRI and NotI (lane 3). (**C**) SDS-PAGE analysis of recombinant calreticulin expression in methanol-induced positive transformants, stained with Coomassie Brilliant Blue: Lane 1—culture supernatant of methanol-induced transformants; Lane 2—flow-through during Ni-NTA affinity purification; Lane 3—wash fraction; Lane 4—eluted protein with 200 mM imidazole; Lane 5—desalted and concentrated protein following ultrafiltration. (**D**) Western blot analysis confirming the identity of purified recombinant goat calreticulin (lane order as in panel (**C**)). (**E**) SEC-HPLC analysis of the molecular weight and purity of yeast-secreted recombinant goat calreticulin. Original Western blot images can be found in Appendix A.

**Figure 4 biomolecules-15-00966-f004:**
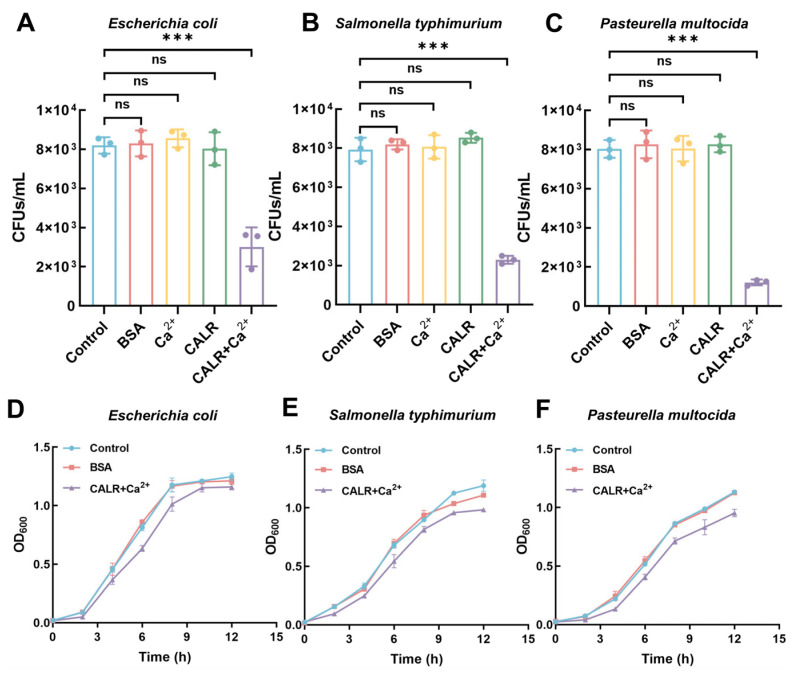
Antibacterial activity of recombinant goat calreticulin. (**A**–**C**) CFU assays were performed to evaluate the inhibitory effects of calreticulin against *Escherichia coli*, *Salmonella typhimurium*, and *Pasteurella multocida*. (**D**–**F**) Growth curve analysis of *Escherichia coli*, *Salmonella typhimurium*, and *Pasteurella multocida* in the presence of calreticulin to assess its impact on bacterial proliferation. All data are presented as mean ± SD from three independent experiments. Statistical significance was determined using one-way ANOVA. ns, no significance; *** *p* < 0.001.

**Figure 5 biomolecules-15-00966-f005:**
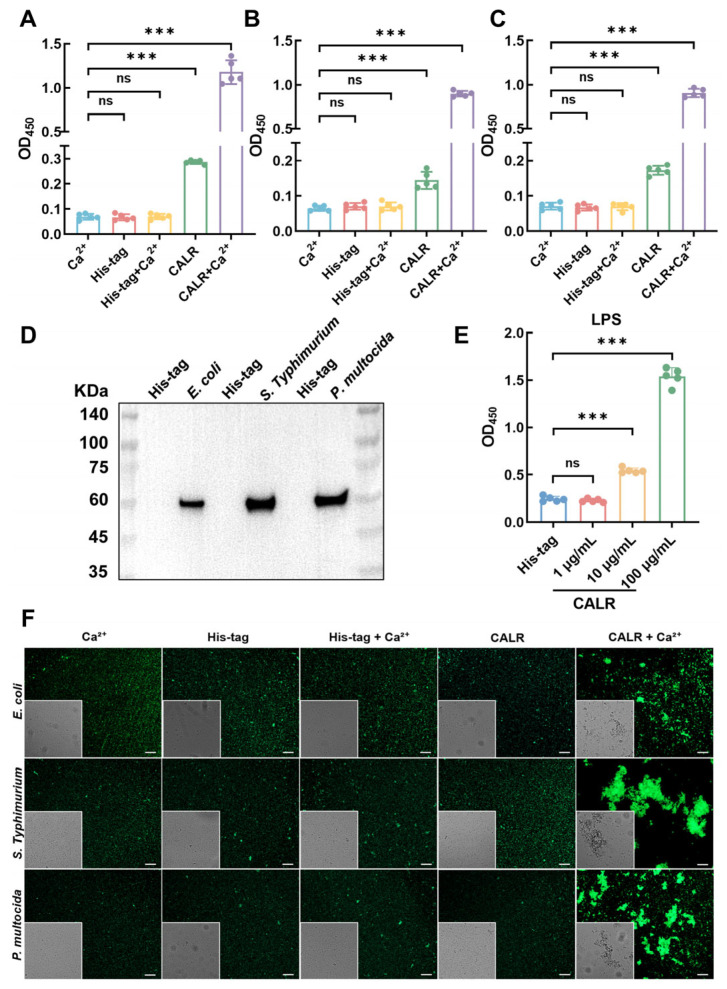
Recombinant goat calreticulin binds to and agglutinates bacteria. (**A**–**C**) ELISA analysis of bacterial binding: *Escherichia coli*, *Salmonella typhimurium*, and *Pasteurella multocida* (1 × 10^7^ CFU/mL) were immobilized on microtiter plates and incubated with either CaCl_2_ (10 mM), His-tag peptide (100 μg/mL), His-tag peptide (100 μg/mL) + CaCl_2_ (10 mM), calreticulin (100 μg/mL), or calreticulin (100 μg/mL) + CaCl_2_ (10 mM). Binding of calreticulin was detected using anti-His tag antibodies. (**D**) Western blot analysis of bacterial pellets after incubation with calreticulin (100 μg/mL) to confirm binding to *Escherichia coli*, *Salmonella typhimurium*, and *Pasteurella multocida*. (**E**) ELISA analysis of calreticulin binding activity to LPS. (**F**) Log-phase *Escherichia coli*, *Salmonella typhimurium*, and *Pasteurella multocida* were labeled with CFSE and incubated with calreticulin (100 μg/mL) ± 10 mM Ca^2+^ at 37 °C for 2 h. Bacterial agglutination was visualized by fluorescence microscopy. Bars = 100 μm. All data are expressed as the mean ± SD from three independent experiments. Statistical significance was assessed using one-way ANOVA. ns, no significance; *** *p* < 0.001. Original Western blot images can be found in Appendix A.

**Figure 6 biomolecules-15-00966-f006:**
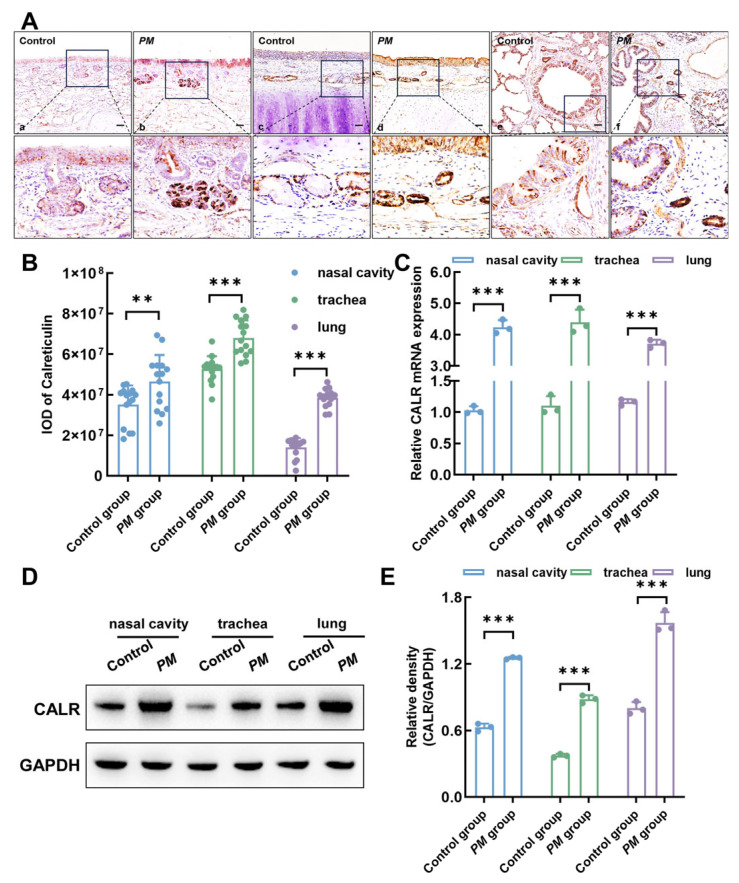
Intranasal infection with *Pasteurella multocida* upregulates calreticulin expression in the respiratory tissues of the lambs. Thirty-day-old lambs were intranasally challenged with *Pasteurella multocida* and euthanized 24 h post-infection, after which respiratory tract tissues, including the nasal cavity, trachea, and lung, were collected for further analysis. (**A**) Immunohistochemical staining showing calreticulin expression in the nasal mucosa, trachea, and lungs following infection. Bars = 20 μm. (**B**) Quantitative analysis of immunohistochemical staining based on gray value measurements. (**C**) RT-qPCR analysis of calreticulin mRNA expression in different respiratory tissues after *Pasteurella multocida* infection. (**D**) Western blot analysis of calreticulin protein levels in infected respiratory tissues. (**E**) Densitometric analysis of Western blot results. All data are presented as mean ± SD from three independent experiments. Statistical significance was evaluated using one-way ANOVA. ** *p* < 0.01; *** *p* < 0.001. Original Western blot images can be found in Appendix A.

**Figure 7 biomolecules-15-00966-f007:**
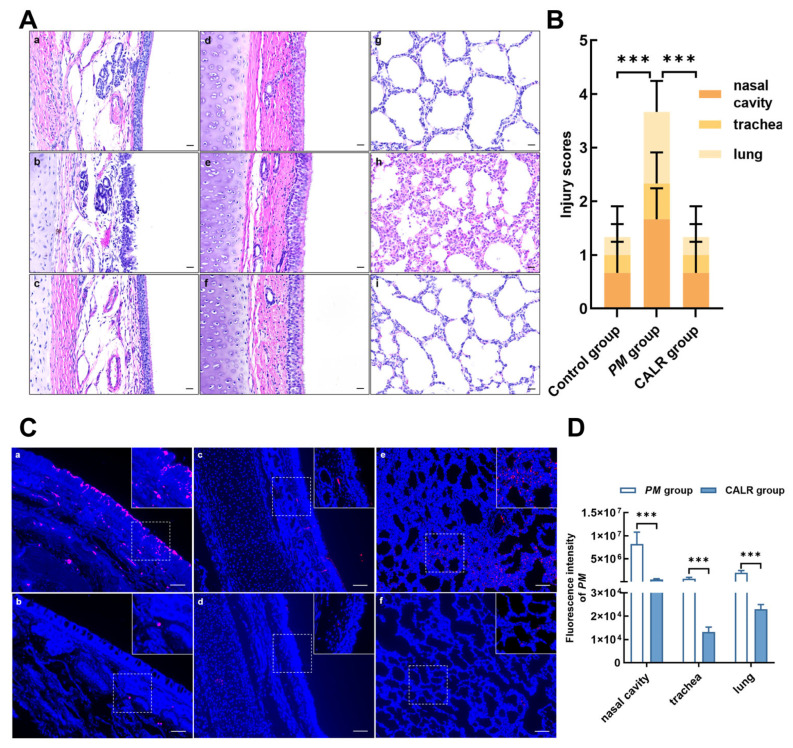
Intranasal administration of calreticulin alleviates *Pasteurella multocida*-induced pathological injury and promotes bacterial clearance in the lambs. Thirty-day-old lambs were intranasally infected with *Pasteurella multocida*; 6 h post-infection, 1 mL of recombinant calreticulin (2.5 mg/mL) was administered intranasally. Lambs were euthanized and necropsied at 24 h post-infection for sample collection. (**A**) Histopathological evaluation of nasal mucosa (**a**–**c**), trachea (**d**–**f**), and lung tissues (**g**–**i**) using H&E staining. Bars = 20 μm. (**B**) Average injury scores calculated for all lambs (*n* = 3 replicates/group). (**C**) In situ hybridization with a *Pasteurella multocida*-specific fluorescent probe to detect bacterial load in nasal cavity (**a**,**b**), tracheal (**c**,**d**), and lung (**e**,**f**) tissue sections. Bars = 100 μm. (**D**) Quantification of fluorescence intensity derived from in situ hybridization results. All data are presented as mean ± SD from three independent experiments. Statistical significance was assessed using one-way ANOVA. *** *p* < 0.001.

## Data Availability

The original contributions presented in this study are included in the article/Appendix A. Further inquiries can be directed to the corresponding author.

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
