# Peer review of "Identification of a Novel Antibacterial Function of Mammalian Calreticulin"

_biomolecules, 2025, doi:10.3390/biom15070966_

Round 1
Reviewer 1 Report
Comments and Suggestions for Authors
In the manuscript “Identification of a Novel Antibacterial Function of Mammalian 2 Calreticulin” by Yichao Ma et al., a calreticulin expression into Pichia pastoris proteomic was carried out. After biochemical and in vitro characterization of the protein and its activity, an in vivo evaluation of the calreticulin protection on mucosa was performed in goats previously infected with Pasteurella multocida.
The document is clear and easy to read. Numerous techniques (WB, ELISA, immunohistology, etc.) were used to produce interesting and relevant data. The figures are clear and useful for understanding the topic. However, a few important remarks need to be answered before publication.
For my point of view, he major problems are (i) the impossibility for the reader to reproduce most of the techniques presented here (see below). They are neither documented nor referenced; (ii) the absence of comparison with another antibacterial molecule, i.e. a positive control. Even if calreticulin has an effect (which is very well demonstrated), it should be compared with another effective drug; (iii) the absence of observation of the effect of calreticulin treatment in living animals.
Major comment
Page 2 lines 59 to 64: The authors emphasize the major interest of the goat because of its high degree of similarity with humans in terms of nasal mucosa, goblet cells, etc. This idea is based on a single reference (ref [21]) which is not relevant. In this publication, the term "goat" appears only once without any connection to the present subject.
Page 3, lines 127-130. Only one dose of calreticulin and calcium was tested. The authors should justify this choice with a reference. A better method would be to test the effect of calreticulin at different doses.
Paragraph 2.6. Typically, ELISA plates are used to immobilize the antibody antigen, not a bacterial culture. Has this test been published before? Is it reproducible? Could the authors add a reference on this? Same goes for LPS. Can the authors also specify the origin of the product (commercial reference)?
Paragraph 2.8. The control with Ca++ only is missing.
Paragraph 2.11. The experimental protocol chose to euthanize all the goats. The authors did not add live animals to observe the effect of calreticulin treatment on the animal after infection, which prevents any conclusions regarding the effectiveness of the drug on the animal. This is a key point. The effectiveness of the treatment on infected goats should have been demonstrated before considering sacrificing them. The experimental design is unacceptable on this point, and the sacrifice of the animals is questionable.
Minor comment
Figure1. The lower part of the table needs to be explained (values below the black cells)
Figure 1 to 6. Some texts should be enlarged
Author Response
|
Comments 1: [Page 2 lines 59 to 64: The authors emphasize the major interest of the goat because of its high degree of similarity with humans in terms of nasal mucosa, goblet cells, etc. This idea is based on a single reference (ref [21]) which is not relevant. In this publication, the term "goat" appears only once without any connection to the present subject.] |
|
Response 1: We sincerely thank the reviewer for highlighting the insufficiency in our literature citation. Upon careful review of the manuscript, we acknowledge that the referenced study does not provide adequate support for our statement regarding the use of goats as a model for human respiratory immune studies. The cited literature primarily discusses the physiological, anatomical, and histological similarities between the respiratory systems of sheep and humans. These include aspects such as respiratory rate, bronchial architecture, arterial blood supply to the bronchi, and the proportion of ciliated cells and goblet cells in the airway epithelium. Owing to these similarities, sheep have been used as animal models in the study of respiratory diseases such as asthma, chronic obstructive pulmonary disease (COPD), and cystic fibrosis, thereby providing an important foundation for modeling human airway immune responses. Given that goats and sheep both belong to the subfamily Caprinae within the family Bovidae, they share high structural and immunological resemblance, particularly in the architecture of the upper respiratory mucosa and the distribution of immune cells. In our original manuscript, the citation was intended to indirectly support the rationale for employing goats in nasal mucosa immunity research. To strengthen this argument, we have now included three additional authoritative and highly relevant references in the revised manuscript, which further support the use of goats as a model from the perspectives of anatomical structure, mucosal immune function, and respiratory epithelial composition. In response to the reviewer’s valuable suggestion, we have also revised the manuscript wording to more accurately reflect the literature, expanding the model description from "goat" to "small ruminants (e.g., sheep and goats)" (page 2, lines 78–84). The newly cited references are now listed in the revised manuscript (page 21, lines 672–677; References [25–27]), and they collectively provide a more robust, systematic rationale for the use of goats in respiratory mucosal immunology studies. We are grateful for the reviewer’s insightful feedback, which has significantly improved the clarity, accuracy, and scientific rigor of our manuscript. |
|
Comments 2: [Page 3, lines 127-130. Only one dose of calreticulin and calcium was tested. The authors should justify this choice with a reference. A better method would be to test the effect of calreticulin at different doses.] |
|
Response 2: We sincerely thank the reviewer for the valuable and constructive comments regarding the experimental dose settings. In response to the concern that “only a single dose of calreticulin and calcium ions was used (Page 3, lines 127–130),” we have made the following detailed clarifications and modifications in the revised manuscript: First, we have added a clarification in the Materials and Methods section (page 4, lines 150–152) indicating that the final concentration of calreticulin used in this study was 100 μg/mL, and the calcium ion concentration was 10 mM. This dosage range was rationally selected based on established experimental conditions commonly applied in previous studies evaluating the antibacterial function of calreticulin in aquatic species. For example, Huang et al. (Molecular Immunology, 2021) demonstrated that 100 μg/mL recombinant calreticulin combined with 10 mM Ca²⁺ exhibited effective bacteriostatic activity in the Takifugu obscurus model. Similar results were also reported by Sun et al. (Fish & Shellfish Immunology, 2023) under identical conditions. Therefore, we adopted this dosage as a foundational reference for validation in our mammalian model (goats). We have now included these references and the rationale for dosage selection in the revised manuscript. Second, in response to the reviewer’s suggestion, we have conducted additional dose-gradient experiments using increasing concentrations of calreticulin (revised Figure S3). The results demonstrated a dose-dependent reduction in bacterial colony-forming units (CFUs) of Escherichia coli, Salmonella typhimurium, and Pasteurella multocida under calcium-supplemented conditions. Specifically, treatment with 1000 μg/mL of calreticulin led to CFU reductions of approximately 1.69 ± 0.07 log, 2.61 ± 0.20 log, and 2.29 ± 0.22 log for the three bacterial strains, respectively. A similar concentration-responsive trend was observed in bacterial growth curves, with higher calreticulin concentrations delaying bacterial proliferation more markedly (page 11, lines 367–380). We are grateful for the reviewer’s insightful feedback, which has strengthened the methodological rigor and scientific validity of our study. Comments 3: [Paragraph 2.6. Typically, ELISA plates are used to immobilize the antibody antigen, not a bacterial culture. Has this test been published before? Is it reproducible? Could the authors add a reference on this? Same goes for LPS. Can the authors also specify the origin of the product (commercial reference)?] Response 3: We sincerely thank the reviewer for the critical and constructive comments regarding the design and reproducibility of the ELISA assay described in Section 2.6. In response to the concerns about the scientific rationale, literature support, and reagent sourcing for coating bacteria or LPS onto ELISA plates, we have carefully reviewed the relevant methodology and made the following detailed revisions and clarifications in the revised manuscript: On the rationale and literature support for coating bacteria or LPS in ELISA assays: As added in Section 2.6 of the revised Materials and Methods, while traditional ELISA often involves coating purified antigens or antibodies, several published studies have validated the use of high-binding ELISA plates (e.g., Costar 3590) for directly coating whole bacterial cells or pathogen-associated molecular patterns (PAMPs) to assess protein–pathogen interactions. This method has been shown to offer good reproducibility and specificity. For example: Huang et al. (Fish & Shellfish Immunology, 2019) evaluated the binding of recombinant PGN recognition protein from Hyriopsis cumingii to seven bacterial strains, LPS, and PGN. In another study by Mu et al. (Journal of immunology, 2020), the authors coated Streptococcus agalactiae, Aeromonas hydrophila, and mannan onto 96-well plates to assess the binding activity of Oreochromis niloticus mannose-binding lectin. Wu et al. (Frontiers in immunology, 2020) analyzed the binding of complement C3 fragments from Paralichthys olivaceus to eight bacterial species using a similar approach, with bacteria coated directly onto ELISA plates. These studies collectively support the validity and reproducibility of using intact bacteria or bacterial components as coating agents in ELISA-based protein-binding assays. To further substantiate our methodology, we have cited these three references in the revised manuscript (page 21, lines 681–687; References [30–32]). On the source of LPS used in the experiment: In response to the reviewer’s suggestion, we have now included detailed information regarding the source and catalog number of the LPS used. Specifically, the LPS was derived from Escherichia coli O55:B5 and purchased from Sigma-Aldrich (Catalog No. L6529). The relevant details have been clearly specified in the revised Materials and Methods section (page 4, lines 169–171). Comments 4: [Paragraph 2.8. The control with Ca+ only is missing.] Response 4: We sincerely thank the reviewer for the important and constructive comments regarding the experimental design described in Section 2.8. Concerning the issue that “a control group with Ca²⁺ alone was not included”, we have carefully reviewed our original experimental protocol, data presentation, and figure legends. We confirmed that a group treated with Ca²⁺ alone was indeed included in the experiment. However, due to insufficient clarity in the original manuscript, this group was not explicitly recognizable to readers. We apologize for this oversight. In response to the reviewer’s suggestion and to improve the clarity of the manuscript, we have revised Section 2.8 of the Materials and Methods (page 5, lines 209–215). Specifically, we now explicitly describe the five experimental groups as follows: (1) Ca²⁺ only group: To control for the effect of calcium ions alone on bacterial growth; (2) His-tag group: To assess any potential effects of the recombinant protein tag itself; (3) His-tag + Ca²⁺ group: To exclude possible non-specific effects of the His-tagged protein in the presence of calcium; (4) Calreticulin group: To evaluate the function of calreticulin in the absence of Ca²⁺; (5) Calreticulin + Ca²⁺ group: To examine the bacterial agglutination activity of calreticulin in the presence of Ca²⁺. We have also enlarged the font size of group labels to ensure that the experimental design is logically presented and easily understood by readers. We appreciate the reviewer’s comment, which has helped us significantly improve the clarity and rigor of our manuscript. Comments 5: [Paragraph 2.11. The experimental protocol chose to euthanize all the goats. The authors did not add live animals to observe the effect of calreticulin treatment on the animal after infection, which prevents any conclusions regarding the effectiveness of the drug on the animal. This is a key point. The effectiveness of the treatment on infected goats should have been demonstrated before considering sacrificing them. The experimental design is unacceptable on this point, and the sacrifice of the animals is questionable.] Response 5: We sincerely appreciate the reviewer’s professional and constructive comments regarding the experimental animal handling strategy described in Section 2.11. This study employed an acute infection model of Pasteurella multocida, a pathogen known for its high virulence in goats. Within 24 h post-infection, animals in the infected group developed obvious clinical signs, including high fever (>40.5°C), mental depression, anorexia, and rapid respiration. Some individuals also presented with more severe symptoms, such as mucosal congestion and respiratory distress. In consideration of animal welfare and in strict compliance with ethical guidelines, humane euthanasia was performed in accordance with the approved protocol by the Institutional Animal Care and Use Committee (IACUC) when animals displayed persistent distress or irreversible pathological changes. This humane endpoint was implemented to prevent further suffering and to obtain high-quality nasal tissue samples for subsequent analyses. Therefore, the termination of the experiment was based on ethical considerations and actual pathological manifestations, rather than a premature decision due to the absence of observable therapeutic effects. In response to the reviewer’s concern that “the absence of in vivo monitoring data limits the conclusion regarding the therapeutic effect of calreticulin”, we have made the following clarifications and revisions in the manuscript: First, in the revised Materials and Methods (Section 2.11, page 6, lines 246–255), we have provided a detailed explanation of the humane endpoint criteria, including the specific clinical symptoms observed in infected goats. Second, in the Discussion section (pages 19, lines 588–598), we have acknowledged the current limitation related to the lack of longitudinal in vivo data. We have emphasized that the therapeutic potential of calreticulin in improving clinical signs and promoting recovery requires further validation in live-animal studies. To address this limitation, we plan to conduct follow-up in vivo intervention studies using Pasteurella multocida infection models of varying severity (low, medium, and high doses) to simulate a range of clinical conditions. These experiments will include continuous monitoring of body temperature, body weight, clinical scoring, respiratory rate, nasal secretions, and bacterial burden in nasal tissues. In addition, we will perform immunohistochemical analysis and cytokine profiling to comprehensively evaluate the therapeutic efficacy and safety of calreticulin. The above revisions have been incorporated into the revised manuscript, and the proposed in vivo validation studies will be a major focus of our future work. Once again, we are sincerely grateful to the reviewer for raising this critical point, which has helped us refine our experimental design and clarify the next steps in our research. Comments 6: [Figure1. The lower part of the table needs to be explained (values below the black cells).] Response 6: Thank you for your valuable comment regarding the completeness of the data presented in Figure 1B. Figure 1B displays a bidirectional alignment matrix of calreticulin amino acid sequences. In this figure, the upper triangular region of the table shows the percentage of sequence identity (% identity), while the lower triangular region indicates the percentage of sequence divergence (% divergence) between pairs of sequences. The black squares represent self-alignments (i.e., alignment of each sequence with itself), with the corresponding divergence values being 0. To improve the clarity and accuracy of the figure, we have revised the figure legend (pages 8, lines 300–304) to explicitly describe these details and clearly distinguish the biological significance represented by the upper and lower triangular regions. Comments 7: [Figure 1 to 6. Some texts should be enlarged.] Response 7: We appreciate the reviewer’s valuable comment regarding the font size used in figure annotations. We acknowledge that in the original submission, certain text elements—including figure legends, axis labels, and group identifiers—were indeed presented in relatively small font sizes, which may hinder clear interpretation by readers, particularly when figures are scaled down or printed. In response, we have conducted a comprehensive review and formatting adjustment of all textual components in Figures 1 through 6. We hope these improvements will enhance the overall visual quality and scientific clarity of the figures. |
|
4. Response to Comments on the Quality of English Language |
|
Point 1: [The English is fine and does not require any improvement.] |
|
Response 1: We sincerely thank the reviewer for the positive comment and are glad to know that the English language of our manuscript is considered clear and does not require any improvement. |
|
5. Additional clarifications |
|
We sincerely thank the reviewer for their careful evaluation; we have no additional clarifications at this time and greatly appreciate the constructive feedback provided. |
Reviewer 2 Report
Comments and Suggestions for Authors
Abstract
Line 10: the authors do not mention any assumption why it was necessary or interesting to analyse the antimicrobial activity of calreticulin. the authors should explain the motivations of the work already at this stage in the light of evidence.
In general, the abstract is more a summary of the results instead of an overall description of the rationale.
Introduction
Even in the introduction the background and rationale is not well defined.
Methods
2.6 Calreticulin-bacteria binding assays. Authors does not show any reference of these assays (ELISA assay and LPS binding assay), they should to demonstrate the robustness of such assay as they do.
Results
Line 236 -244: how authors could state this?
“Figure 1A demonstrated that calreticulin is highly conserved among mammals, suggesting its fundamental role in antibacterial immunity …. indicating its conserved and crucial function in maintaining basic antibacterial defense mechanisms”.
The high protein sequence conservation does not necessary relate with a putative role in “antibacterial immunity” whatever it means.
Authors should better clarify this aspect.
3.4. Verification of antibacterial activity of calreticulin.
Authors do not provide any explanation of antimicrobial activity of calreticulin exclusively in presence of calcium nor in results nor in discussion. Why the authors suppose that this protein in antimicrobial protein? And how it works?
Data presented for authors does not highlight a surprising antimicrobial activity of the protein since the bacterial reduction is from 8 x10^3 until to 2 x 10^3. even the grow curve do not show a significative bacterial decrease. Authors should consider higher protein concentration to state that it is antimicrobial and the activity is dose-dependent.
Line 327: “3.5. Calreticulin binds to and aggregates bacteria” Correct the title to increase readability
Figure 5
What LPS have been used for the experiments?
Authors expressed the protein binding as OD450, but to improve reproducibility and give a more informative data they should use a calibration to identify the amount of protein that bind the bacteria.
Author Response
|
Comments 1: [Line 10: the authors do not mention any assumption why it was necessary or interesting to analyse the antimicrobial activity of calreticulin. the authors should explain the motivations of the work already at this stage in the light of evidence.] |
|
Response 1: We sincerely appreciate the reviewer’s constructive comment regarding the insufficient expression of the research motivation in the Abstract. We fully acknowledge the concern raised and have accordingly made substantial content and logical revisions to the introductory section of the Abstract to more clearly present the study background and scientific hypothesis. Specifically, previous studies have demonstrated that calreticulin not only functions as a classical molecular chaperone and immune regulator, but also exhibits potential antibacterial activity in aquatic animals—for example, by binding bacterial and pathogen-associated molecular patterns (PAMPs) to participate in the antibacterial immune response in fish and shellfish. However, whether calreticulin possesses similar antibacterial functions in mammals remains largely unexplored, and systematic evidence is still lacking. On this basis, the present study employed goats as an experimental model to provide a comprehensive multi-level evaluation of calreticulin’s antibacterial effects, including recombinant protein expression, in vitro antibacterial activity assays, binding capability analyses, and an in vivo nasal infection model. This approach aims to elucidate the antibacterial role of calreticulin in mammals and to offer a theoretical foundation for host-targeted anti-infective strategies. We have incorporated this background and research motivation into the opening section of the Abstract (page 1, lines 9–13), thereby enhancing the rational structure and logical coherence of the Abstract, enabling readers to fully grasp the scientific question and core hypothesis of the study at the earliest stage. |
|
Comments 2: [In general, the abstract is more a summary of the results instead of an overall description of the rationale.] |
|
Response 2: We sincerely appreciate the reviewer’s constructive evaluation of the Abstract’s structure and logical expression. We fully understand the reviewer’s concern that the original Abstract primarily focused on the presentation of experimental results but did not adequately convey the study’s background rationale, theoretical hypothesis, and scientific motivation. This limitation hindered readers from gaining a comprehensive initial understanding of the overall framework and academic significance of the research. To address this issue, we have systematically revised and restructured the Abstract (all the revisions have been clearly marked in red in the revised manuscript for ease of reference), mainly encompassing the following aspects: (1) Supplementing research motivation and theoretical basis: We added content at the beginning of the Abstract to emphasize that calreticulin, as a classical molecular chaperone, has been identified to possess potential antibacterial functions in aquatic animals. However, there is currently no systematic evidence confirming whether calreticulin exerts similar antibacterial effects in mammals, representing an important scientific question that urgently requires investigation. (2) Clarifying research objectives and significance: Subsequently, we clarified that the present study aims to systematically verify the antibacterial function of calreticulin using a goat model and explore its antibacterial potential in infection response. This work seeks to fill the existing knowledge gap and advance the potential translational application of calreticulin as an antibacterial agent in both veterinary and human medicine. (3) Optimizing logical structure and paragraph cohesion: We reconstructed the logical flow of the Abstract by organizing it along the main line of “problem—hypothesis—methods—results—significance.” Additionally, necessary transitional sentences were incorporated between sections to enhance clarity of the research rationale, improve content cohesion, and strengthen the overall logical rigor. Comments 3: [Even in the introduction the background and rationale is not well defined.] Response 3: We sincerely thank you for your careful review and valuable suggestions on our manuscript. In response to your concern regarding the insufficient elaboration of the background and research motivation in the Introduction, we have made the following revisions to enhance the logical flow and scientific rigor of the manuscript (all the revisions have been clearly marked in red in the revised manuscript for ease of reference): (1) Strengthening the systematic presentation of research background: We conducted a more comprehensive literature review, particularly focusing on existing studies of calreticulin’s antibacterial functions, highlighting recent discoveries in aquatic organisms and the current lack of research in mammals. Additionally, we supplemented the Introduction with a discussion on the global challenge of antibiotic resistance, emphasizing the urgent need for novel anti-infective strategies and the advantages of host protein-based therapies, thereby clarifying the practical significance and scientific value of our study. (2) Clarifying research motivation and scientific basis: Building on the above background, we explicitly stated that this study aims to fill the knowledge gap concerning the antibacterial function of calreticulin in mammals and to investigate its potential antibacterial role against Pasteurella multocida respiratory infection. We also explained the scientific rationale for choosing goats as the animal model, including their respiratory anatomy and immune mechanisms that are comparable to humans, thus enhancing the translational relevance and clinical value of the research. (3) Revising the introduction structure to enhance logical coherence: We optimized paragraph transitions to ensure a clear, progressive narrative from background introduction, problem statement, research necessity, to study objectives, achieving rigorous and natural logical flow. Comments 4: [2.6 Calreticulin-bacteria binding assays. Authors does not show any reference of these assays (ELISA assay and LPS binding assay), they should to demonstrate the robustness of such assay as they do.] Response 4: We sincerely thank the reviewer for the insightful comments and concerns regarding the scientific validity and reproducibility of the experimental method described in Section 2.6 “Calreticulin-bacteria binding assays.” To address this, we have added a detailed explanation of the experimental rationale and relevant literature support in the revised Section 2.6 of the Materials and Methods. Although traditional ELISA is commonly used for detecting antigen–antibody interactions, recent studies have increasingly applied ELISA–based approaches in analyzing interactions between immune–related proteins and pathogens. These approaches typically involve coating ELISA plates with fixed bacteria, LPS, or PGN to assess the binding capacity of specific proteins. Such methods have demonstrated excellent reproducibility and have been validated in multiple model organisms, including fish and mollusks. For example: Huang et al. (Fish & Shellfish Immunology, 2019) employed ELISA plates coated with seven types of bacteria, as well as LPS and PGN, to evaluate the binding activity of a peptidoglycan recognition protein from Hyriopsis cumingii. Mu et al. (Journal of immunology, 2020) used a mannose-binding lectin from Oreochromis niloticus in binding assays with Aeromonas hydrophila and Streptococcus agalactiae, assessed via mannose-coated ELISA. Wu et al. (Frontiers in immunology, 2020) utilized a similar method to assess the binding of complement C3 fragments from Paralichthys olivaceus to eight bacterial species. To further substantiate our methodology, we have cited these three references in the revised manuscript (page 4, lines 169–171; References [30–32]). Comments 5: [Line 236-244: how authors could state this? “Figure 1A demonstrated that calreticulin is highly conserved among mammals, suggesting its fundamental role in antibacterial immunity …. indicating its conserved and crucial function in maintaining basic antibacterial defense mechanisms”.] Response 5: We sincerely thank the reviewer for the careful evaluation of our manuscript and for the valuable comments. We acknowledge that the original description of Figure 1A (the phylogenetic tree of calreticulin amino acid sequences) lacked sufficient precision and logical clarity, failing to adequately present the scientific basis linking the phylogenetic results to the study’s interpretations. We apologize for this oversight. In response to your suggestion, we have revised and expanded the relevant paragraphs to improve scientific clarity and logical coherence, as follows: (1) Clarifying the scientific meaning and purpose of Figure 1A: In the Results section, we have explicitly stated that Figure 1A presents a phylogenetic tree constructed based on calreticulin amino acid sequences from 40 species. The results indicate that calreticulin sequences among mammals are highly conserved, as reflected by their close evolutionary distances, tight clustering, and high bootstrap support for most branches. This suggests that calreticulin may play a conserved and fundamental physiological role across mammalian species. A detailed explanation of this figure has been provided in the revised text (pages 6–7, lines 274–287). (2) Removing unsupported interpretative statements: We have removed speculative phrases in the original version—such as “suggesting its fundamental role in antibacterial immunity”—that lacked direct experimental evidence. To enhance scientific rigor, we avoided equating evolutionary conservation with antibacterial function in the absence of supporting data. Comments 6: [The high protein sequence conservation does not necessary relate with a putative role in “antibacterial immunity” whatever it means. Authors should better clarify this aspect.] Response 6: We sincerely thank the reviewer for the careful assessment of our work and for the constructive suggestions. We fully agree with your comment that “high sequence conservation of a protein does not necessarily imply a functional role in antibacterial immunity,” and we apologize for the lack of rigor in our original wording, which may have led to misinterpretation. In response to your concern, we have made the following specific revisions to improve the scientific accuracy and logical reasoning in this section (page 7, lines 291–294): (1) Removal of overstated inferences: We acknowledge that the original phrasing—such as “Taken together, calreticulin exhibits strong sequence conservation and evolutionary stability across species, indicating its conserved and crucial function in maintaining basic antibacterial defense mechanisms”—inappropriately inferred a functional role based solely on evolutionary and sequence conservation. To address this, we have deleted such statements from the revised manuscript. (2) Addition of more accurate and balanced interpretation: In the revised version, we have modified the relevant sentence to: “This strong sequence conservation and evolutionary stability support the notion that calreticulin may carry evolutionarily conserved physiological functions that have been stably maintained throughout evolution. However, its potential role in antibacterial defense requires further experimental validation.” Comments 7: [3.4. Verification of antibacterial activity of calreticulin. Authors do not provide any explanation of antimicrobial activity of calreticulin exclusively in presence of calcium nor in results nor in discussion. Why the authors suppose that this protein in antimicrobial protein? And how it works?] Response 7: We sincerely thank the reviewer for the thorough evaluation of our work and for the valuable suggestions. We highly appreciate your observation regarding the lack of explanation in the Results and Discussion sections on why calreticulin exhibits antibacterial activity only in the presence of calcium ions, as well as the rationale for attributing antibacterial function to this protein and its potential mechanisms of action. We take this concern very seriously and have revised and supplemented the relevant content accordingly (page 18, lines 553–569). In the Discussion section, we have added a new paragraph explaining that the calcium-dependent antibacterial activity of calreticulin is likely related to its calcium-regulated conformational changes and activation mechanisms. Calreticulin is a high-affinity calcium-binding chaperone, with a C-terminal domain enriched in aspartic acid and glutamic acid residues capable of coordinating multiple Ca²⁺ ions. Upon calcium binding, calreticulin undergoes significant conformational changes, shifting from a flexible " folding intermediate" state to a stable three-dimensional structure. This structural transition is believed to be essential for its ability to bind various molecular targets, including pathogen-associated molecular patterns (PAMPs). Furthermore, the antibacterial effect of calreticulin is not limited to molecular binding but also involves the induction of bacterial aggregation, which reduces bacterial motility. This aggregation physically restricts bacterial movement and dissemination, and may further enhance antimicrobial action through multiple mechanisms, including electrostatic shielding, structural stabilization, and increased hydrophobic interactions at the protein–bacteria interface. Collectively, these effects may disrupt bacterial membrane homeostasis, thereby inhibiting bacterial growth. Comments 8: [Data presented for authors does not highlight a surprising antimicrobial activity of the protein since the bacterial reduction is from 8 x10^3 until to 2 x 10^3. even the grow curve do not show a significative bacterial decrease. Authors should consider higher protein concentration to state that it is antimicrobial and the activity is dose-dependent.] Response 8: We sincerely thank the reviewer for the in-depth evaluation of our work and for the valuable comments regarding the antibacterial activity results. In response to the suggestion, we performed additional dose-gradient experiments using higher concentrations of recombinant calreticulin (500 μg/mL and 1000 μg/mL) to further investigate its antibacterial properties. The results showed bacterial survival declined progressively with increasing calreticulin concentration (Figure S3A-3C). Specifically, treatment with 1000 μg/mL of calreticulin led to a reduction of CFUs by approximately 1.69 ± 0.07 log for Escherichia coli, 2.61 ± 0.20 log for Salmonella typhimurium, and 2.29 ± 0.22 log for Pasteurella multocida compared to the untreated control group. This concentration-dependent bacteriostatic effect was also supported by bacterial growth curve analyses, where increasing concentrations of calreticulin were associated with delayed bacterial proliferation (Figure S3D-3F) (page 11, lines 367–380). These findings provide preliminary evidence that calreticulin exhibits dose-dependent antibacterial activity in vitro. However, we recognize that the observed activity was not robust at lower concentrations, and thus we have revised the original manuscript to avoid conclusive terms such as “significant antibacterial activity”. Instead, we now describe these effects as preliminary findings and acknowledge the limitations, emphasizing that further mechanistic investigation is needed. It is worth noting that, although the in vitro bactericidal effect appears limited, calreticulin demonstrated stronger anti-infective efficacy in vivo. In a nasal infection model using lambs challenged with Pasteurella multocida, intranasal supplementation with recombinant calreticulin significantly reduced bacterial burden and alleviated tissue pathology. These protective effects are likely not solely attributable to its direct bactericidal activity but may also involve its immunomodulatory functions. Previous studies have reported that calreticulin can act as an immunological adjuvant by promoting dendritic cell maturation, enhancing antigen presentation, and subsequently activating antigen-specific T cell responses [1]. Such adjuvant-like effects may also contribute to enhanced pathogen clearance in the infection context by activating both innate and adaptive immune responses. Moreover, calreticulin has been shown to enhance macrophage phagocytosis. Upon binding to bacterial surface components such as LPS or other pathogen-associated molecular patterns (PAMPs), calreticulin can facilitate uptake and killing by phagocytic cells like macrophages [2]. Therefore, in the revised manuscript, we have carefully revised or removed statements such as “significant antibacterial activity”, and replaced them with more cautious expressions such as “preliminary antibacterial capacity” to objectively reflect the current level of evidence. Future studies will aim to elucidate how calreticulin coordinates innate and adaptive immune responses in vivo, further clarifying its role and therapeutic potential in antibacterial immune defense. 1. Gong, Z., Chen, M., Miao, J., Han, C. J., Zhong, Q., Gong, F. Y., & Gao, X. M. Calreticulin as an adjuvant in vivo to promote dendritic cell maturation and enhance antigen-specific T lymphocyte responses against melanoma. J. Immunol. Res. 2022, 2022, 8802004. 2. Sun, J. Q.; Zhao, K. Y.; Zhang, Z. X.; Li, X. P. Two novel teleost calreticulins PoCrt-1/2, with bacterial binding and agglutination activity, are involved in antibacterial immunity. Fish. Shellfish. Immunol. 2023, 143, 109203. Comments 9: [Line 327: “3.5. Calreticulin binds to and aggregates bacteria” Correct the title to increase readability.] Response 9: We sincerely thank the reviewer for the valuable suggestion regarding the section title. In response, we have revised the title to “Calreticulin mediates bacterial binding and aggregation” to more accurately reflect the functional role of calreticulin in binding to bacteria and promoting their aggregation, thereby improving both the scientific precision and readability of the manuscript. Comments 10: [Figure 5: What LPS have been used for the experiments? Authors expressed the protein binding as OD450, but to improve reproducibility and give a more informative data they should use a calibration to identify the amount of protein that bind the bacteria.] Response 10: We sincerely thank the reviewer for the valuable comments regarding our experimental methods and data presentation. In response to your concerns about the unspecified source of LPS and the lack of quantifiability and reproducibility in OD450-based protein binding measurements, we have made the following revisions and improvements: (1) Clarification of LPS source and type: We have now included detailed information regarding the LPS used in our experiments. Specifically, the LPS was purchased from Sigma-Aldrich (catalog number: L6529), derived from Escherichia coli O55:B5. This information has been added to the “Materials and Methods” section (page 4, lines 169–171). (2) Improvement of protein-binding quantification to enhance reproducibility and clarity: We fully agree with the reviewer’s concern that using OD450 values alone to represent protein-binding capacity may lack clarity and quantitativeness. In the revised manuscript, we have optimized our approach as follows: We conducted bacterial binding assays using a range of recombinant calreticulin concentrations (e.g., 1, 10, 100, 500, and 1000 μg/mL) to assess dose dependence and potential binding saturation (page 12, lines 394–405). This modified design allows a more intuitive evaluation of the interaction between calreticulin and bacteria, and lays a foundation for future dose–response mechanism studies. |
|
4. Response to Comments on the Quality of English Language |
|
Point 1: The English could be improved to more clearly express the research. |
|
Response 1: We sincerely thank the reviewer for the constructive suggestion regarding the clarity of our English expression. In response, we have carefully proofread and revised the entire manuscript to improve language accuracy, readability, and scientific clarity. The revised version has undergone comprehensive English editing to ensure that the research rationale, methodology, and conclusions are communicated more clearly and professionally. All modified portions have been highlighted in red in the revised manuscript to facilitate your review. |
|
5. Additional clarifications |
|
We sincerely thank the reviewer for their careful evaluation; we have no additional clarifications at this time and greatly appreciate the constructive feedback provided. |
Round 2
Reviewer 1 Report
Comments and Suggestions for Authors
The authors responded satisfactorily to my comments and corrected the article, which is now ready for publication.
Reviewer 2 Report
Comments and Suggestions for Authors
In my opinion the article is now worthy of publication.